# Quasi-Static Lineshape Theory for Rydberg Excitations in High-Density Media

Trevor Scheuing [1,2] and Jesús Pérez-Ríos [3,4,*]

1 Department of Physics, Hamilton College, Clinton, NY 13323, USA; tscheuin@hamilton.edu
2 Department of Mathematics, Syracuse University, Syracuse, NY 13244, USA
3 Department of Physics and Astronomy, Stony Brook University, Stony Brook, NY 11794, USA
4 Institute for Advanced Computational Science, Stony Brook University, Stony Brook, NY 11794, USA
* Correspondence: jesus.perezrios@stonybrook.edu

**Abstract:** This work presents a theoretical approach for lineshapes of Rydberg excitations in high-density media. In particular, we introduce the quasi-static lineshape theory, leading to a methodic and general approach, and its validity is studied. Next, using $^{84}$Sr as a prototypical scenario, we discuss the role of the thermal atoms and core–perturber interactions, generally disregarded in Rydberg physics. Finally, we present a characterization of the role of Rydberg–core perturber interactions based on the density and principal quantum number that, beyond affecting the lineshape, could potentially apply to chemi-ionization reactions responsible for the decay of Rydberg atoms in high-density media.

**Keywords:** Rydberg excitations; absorption spectrum; lineshape; quasi-static; Monte Carlo

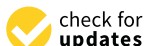



## 1. Introduction

The pioneering work of Amaldi and Segrè about the spectroscopy of Rydberg atoms in high-density media showed an unexpected density-dependent shift of the lines [1]. Fermi explained this shift as a consequence of the Rydberg electron's scattering with the perturbers within the Rydberg orbit [2]. In the ultracold regime, the lack of significant thermal fluctuations enables attractive electron–perturber interactions to bind perturbers to the Rydberg core, forming ultra-long-range Rydberg molecules [3–7]. Despite being homonuclear, these molecules show a dipole moment [8–17]. Similarly, the possibility of having Rydberg–perturber bound states gives rise to interesting many-body effects known as Rydberg polarons [18,19], in which the Rydberg excitation is dressed with those bound states of the background gas. On the other hand, electron–perturber interactions induce the Rydberg to decay faster than in a vacuum due to l-changing collisions and chemical-ionization reactions [20–23].

Rydberg's properties in high-density media are encoded in the excitation spectrum [18,24,25]. Therefore, it is one of the most relevant tools to diagnose Rydberg properties, characterize electron–neutral interactions, and estimate hard-to-measure physical properties of the background gas, such as the density of the media [26,27]. However, despite its relevance, there is no general theoretical approach to explain the Rydberg excitation lineshape. In the case of the absence of electron–perturber $p$-wave shape resonance, such as in Sr, it is possible to apply many-body methods to get good lineshapes [18,19] in comparison with experimental observations. On the contrary, when such a resonance exists, a quasi-static approach for the lineshape leads to a proper description of the Rydberg excitation lineshape [25]. Therefore, developing a proper framework for the theory of Rydberg excitation spectra is necessary.

This work develops a general quasi-static lineshape theory for Rydberg excitations in high-density media. In particular, we include effects from thermal and condensate components and analyze the role of the charge-induced dipole interaction in the spectra. Additionally, we study the range of validity of the quasi-static lineshape theory. Thus, we

develop a general framework applicable to Rydberg-background systems. Furthermore, we provide a methodological approach to calculate the lineshape for any Rydberg-background system. The paper organizes as follows: Section 2 introduces the fundamentals of the quasi-static lineshape theory, analyzes its validity, and explains our methodology for Rydberg excitations in high-density media. Section 3 discusses our results, and finally, in Section 4 we state conclusions and outlook of our work. Atomic units are used throughout unless stated otherwise.

## 2. Theoretical Approach and Methodology

### 2.1. Quasi-Static Lineshape Theory

The quasi-static approximation for lineshape is an approach to determining the effect of perturber atoms on the light frequency emitted from a radiating (or absorbing) atom in the limit where the motion of the atoms is negligible during excitation (or absorption) [28]. Kuhn first developed this idea in the context of neutral atom pressure broadening based on the Franck–Condon principle [29–32]. While Kuhn initially limited his approach to the case of a single perturber, Margenau later conducted a statistical calculation to show that the same limit applies in the case of multiple perturbers [33–36].

In particular, let $E_i$ and $E_f$ be the initial and final energies, respectively, of an atom due to the absorption of a photon. In the absence of perturbers, the absorption frequency is $\omega_0 = (E_f - E_i)$. If we introduce perturbers but all atoms are sufficiently slow (they move negligibly during the excitation or emission time), then the new initial and final energies $E_i + \Delta E_i$ and $E_f + \Delta E_f$, respectively, are approximately constant during the emission of the photon. Thus, the new emission frequency is (in atomic units) [28]

$$\omega = (E_f - E_i) + (\Delta E_f - \Delta E_i) = \omega_0 + \Delta\omega \tag{1}$$

where $\Delta\omega = \Delta E_f - \Delta E_i$. We call the quantity $\Delta\omega$ the detuning, which represents the change in absorption frequency due to the presence of perturbers.

The quasi-static approach is applicable as long as the excitation time, $\tau_{ex} = |\Delta\omega|^{-1}$, is much shorter than the collision time between the perturber and the emitter (or observer) $\tau_{col}$. In the case of a Rydberg excitation in a background gas, the collision time can be calculated as $\tau_{col} = \frac{b}{\langle v \rangle}$, where $b$ stands for the impact parameter, which we take as equal to the size of the Rydberg orbit, $b = 2n^{*2}$, where $n^*$ denotes the effective principal quantum number. The average perturber velocity is given by $\langle v \rangle = \sqrt{\frac{8k_B T}{\pi m}}$ where $k_B$ is the Boltzmann's constant, $T$ is the temperature, and $m$ is the mass of the perturber. Therefore, the quasi-static approximation applies only when $\tau_{ex} \ll \tau_{col}$, or

$$|\Delta\omega| \gg \frac{1}{n^{*2}}\sqrt{\frac{2k_B T}{\pi m}} = |\Delta\omega_{min}|. \tag{2}$$

### 2.2. Rydberg Excitation in a Dense Environment

As explained by Fermi [2], the energy of a Rydberg atom in a dense gas is affected by the scattering of the Rydberg electron off perturber atoms within its orbit. In this scenario, the extent of the electronic wavefunction is assumed to be large relative to that of perturber wavefunctions. As a result, the electron–perturber interaction is described by a contact interaction proportional to the electron–perturber scattering length,

$$V_{e-p} = 2\pi a_s(k)\delta^{(3)}(\mathbf{r} - \mathbf{R}), \tag{3}$$

where $k$ is the electron's semiclassical momentum, $a_s(k)$ is the momentum-dependent s-wave electron–perturber scattering length, $\mathbf{r}$ is the position of the electron, $\mathbf{R}$ is the position of the perturber, and $\delta^{(3)}(\mathbf{x})$ is the three-dimensional Dirac delta function of argument $\mathbf{x}$.

However, it is possible to develop this model further by including higher order partial waves on the electron–perturber scattering, as Omont [37] showed via an $l$ expansion of the pseudopotential. In particular, including the $p$-wave partial wave scattering yields

$$V_{e-p} = 2\pi a_s(k)\delta^{(3)}(\mathbf{r} - \mathbf{R}) + 6\pi(a_p(k))^3\delta^{(3)}(\mathbf{r} - \mathbf{R})\overleftarrow{\nabla} \cdot \overrightarrow{\nabla}, \tag{4}$$

where $a_p(k)$ is the $p$-wave momentum-dependent scattering length, and $\overleftarrow{\nabla}$ and $\overrightarrow{\nabla}$ are the left- and right-acting gradients, respectively. The inclusion of $p$-wave effects are essential for systems showing a $p$-wave shape resonance for electron-atom scattering, such as Rb and Cs.

While the electron–perturber interaction is usually the dominant effect, the perturber–core distance may be very short at high enough densities. Therefore, we account for the charge-induced dipole interaction between the positively charged Rydberg core and neutral perturbers given by

$$V_{c-p} = -\frac{\alpha}{2r^4} \tag{5}$$

where $\alpha$ is the dipole polarizability of the perturber.

Consider an excitation of an atom in a dense gas to a Rydberg state. Under the quasi-static approximation, the positions of nearby perturbers are fixed during the excitation. For the initial, ground state, the effect of $V_{e-p} + V_{c-p}$ is negligible. For the target, Rydberg state, according to first order perturbation theory, the excitation energy is affected by $\sum(\langle V_{e-p}\rangle + \langle V_{c-p}\rangle)$, where the sum runs over all perturbers. Although a first order perturbation is not accurate on its own, previous research has shown that it can be made accurate if "effective" scattering lengths that reproduce experimental bound states are substituted for the true scattering lengths [15,38].

### 2.3. Simulating Lineshape of Rydberg Excitation in a Dense Environment

To match experimental setups [18], our simulated Rydberg excitations occur in an atom trap containing a Bose–Einstein condensate (BEC) with the subsequent thermal atomic fraction. The distances $\rho$ of the excited atoms from the center of the trap are selected randomly from the density distribution $\mathcal{N}(\rho) = \mathcal{N}_{BEC}(\rho) + \mathcal{N}_{th}(\rho)$. The trap is assumed to be a spherically symmetric harmonic potential for simplicity, but our method is valid for any trap geometry and kind. Under the Thomas–Fermi approximation, the condensate density distribution can then be shown to be

$$\mathcal{N}_{BEC}(\rho) = \begin{cases} \frac{m^2\omega^2}{8\pi a_{bb}}(R_{TF}^2 - \rho^2) & 0 \le \rho \le R_{TF} \\ 0 & \rho > R_{TF} \end{cases}, \tag{6}$$

where $m$ is the atomic mass, $\omega$ is the frequency of the trap, $a_{bb}$ is the boson–boson scattering length, and $R_{TF}$ is the Thomas–Fermi radius of the trap. For practical purposes, we can rewrite this as

$$\mathcal{N}_{BEC}(\rho) = \begin{cases} \mathcal{N}_{max}\left(1 - \frac{\rho^2}{R_{TF}^2}\right) & 0 \le \rho \le R_{TF} \\ 0 & \rho > R_{TF} \end{cases}, \tag{7}$$

where $\mathcal{N}_{max}$ is the peak BEC density occurring at the center of the trap.

According to Bose–Einstein statistics, the density distribution of a thermal gas of bosons is [39]

$$\mathcal{N}_{th}(\vec{\rho}) = \frac{1}{\lambda_T^3}\mathrm{Li}_{3/2}\left(e^{\frac{-1}{k_BT}(V(\vec{\rho})-\mu)}\right) \tag{8}$$

where $k_B$ is Boltzmann's constant, $T$ is temperature, $\lambda_T = \sqrt{\frac{2\pi}{mk_BT}}$ is the thermal de Broglie wavelength, $V(\vec{\rho})$ is potential energy at position $\vec{\rho}$, $\mu$ is the chemical potential, and $\mathrm{Li}_{3/2}(x)$

is the polylogarithm of order $\frac{3}{2}$ and argument $x$. In this case, including both the external trap potential and the internal interaction potentials yields [19]

$$\mathcal{N}_{th}(\rho) = \frac{1}{\lambda_T^3} \mathrm{Li}_{3/2}\left(e^{\frac{-1}{k_B T}\left(\frac{1}{2}m\omega^2|\rho^2 - R_{TF}^2| + \frac{8\pi a_{bb}}{m}\mathcal{N}_{th}(\rho)\right)}\right), \tag{9}$$

which can be solved numerically for $\mathcal{N}_{th}(\rho)$.

A flow chart describing the steps of our simulation is given in Figure 1. A single iteration proceeds in the following steps:

1.  We choose the distance $\rho$ of the Rydberg excitation from the center of the trap. The probability of a radial distance $\rho$ should be proportional both to the density at $\rho$ and the surface area of a sphere of radius $\rho$. Thus, we use $P(\rho) = C\mathcal{N}(\rho)\rho^2$ as a radial probability distribution where $0 \le \rho \le R_{TF}$ and $C = (\int_0^{R_{TF}} \mathcal{N}(\rho)\rho^2 d\rho)^{-1}$ normalizes the distribution to 1.

2.  We choose the number of nearby perturber atoms $N$. We define "nearby" to mean within $r_{max} = 2.5(n^*)^2$ of the Rydberg core in atomic units, where $n^*$ is the effective principal quantum number of the Rydberg state. Beyond this point, the electronic wavefunction is negligible. We assume $r_{max} \ll R_{TF}$ so that the nearby density is roughly uniform. Thus, the expected number of nearby perturbers is $\lambda = \frac{4}{3}\pi r_{max}^3 \mathcal{N}(\rho)$, and, according to a Poisson distribution, the probability of $N$ nearby perturbers is $P(N) = \frac{\lambda^N e^{-\lambda}}{N!}$.

3.  We choose the distance $r$ of each perturber from the Rydberg core. Since the nearby density is constant, this is a uniform distribution in space, or one proportional to $r^2$. This can be normalized as $P(r) = \frac{3}{r_{max}^3}r^2$ where $0 \le r \le r_{max}$.

4.  The total effect of perturbers on the excitation energy is then $\sum(\langle V_{e-p}\rangle + \langle V_{c-p}\rangle)$, where the sum runs over all perturbers. This gives the detuning for this Rydberg excitation.

After many iterations, a histogram of the resultant detunings would give an approximate lineshape. However, we also simulate the effect of the bandwidth of the light. Each excitation contributes a Lorentzian profile to the total absorption, so the total absorption of a frequency $\nu$ is proportional to [28]

$$I(\nu) = \sum_i \frac{1}{(\nu - \nu_i)^2 + (\Gamma/2)^2} \tag{10}$$

where $i$ runs over all Rydberg excitations (i.e., all iterations), $\nu_i$ is the detuning of the $i^{th}$ excitation, and $\Gamma$ is the bandwidth of the light. Normalizing $I(\nu)$ to 1 gives our final lineshape.

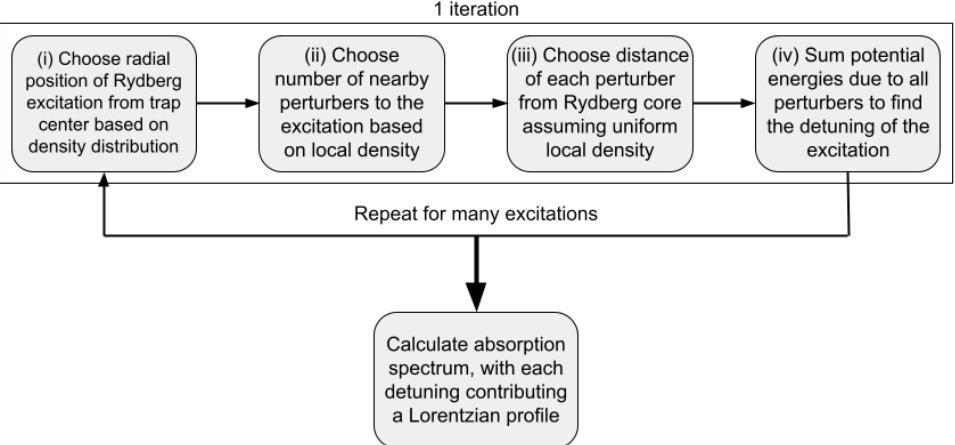

**Figure 1.** A flow chart describing the steps of the simulation. Steps (i)–(iv) are repeated for many iterations. The absorption spectrum is then calculated in the final step.

This computational approach is applicable to BECs in different trap geometries and properties, as long as the size of the Rydberg orbit does not exceed 20% of the $R_{\text{TF}}$, which in the present case is $\sim$2 μm. In other words, our approach is valid as long as the principal quantum number $n \lesssim 120$. Additionally, our approach can be generalized to Rydberg states with angular momentum, but, in that case, one should take into account the orientation of the orbitals in order to compute the lineshape.

## 3. Results and Discussion

In this section, we discuss the quasi-static lineshape approach explained above in the case of a Rydberg excitation in an $^{84}$Sr BEC. In particular, we include the ubiquitous core–perturber interaction that inexorably leads to reliable *s*-wave and *p*-wave scattering lengths for electron–Sr collisions.

### 3.1. Determining Effective Scattering Lengths

To simulate the experimental lineshapes measured by Camargo et al. [18], we calculate the absorption spectra of $^{84}$Sr atoms using the parameters shown in Table 1 for Rydberg excitations to the $49^3$S, $60^3$S, and $72^3$S states.

**Table 1.** Values of relevant parameters. The measured values of $\alpha$ and $a_{bb}$ are taken from Refs. [40,41], respectively.

| $\alpha$ (a.u.) | $m$ (a.u.) | $a_{bb}$ (a$_0$) | $\Gamma$ (MHz) |
|---|---|---|---|
| 186.441 | 153,123 | 123 | 1 |

The *s*-wave and *p*-wave scattering lengths $a_s(k)$ and $a_p(k)$ relevant to the electron–perturber interaction $V_{e-p}$ are not known precisely. Instead, we estimate that $a_s(k) \approx a_s(0) + \frac{\pi}{3}\alpha k$ and $a_p(k) \approx a_p(0)$ [15,38]. The zero momentum limits $a_s(0)$ and $a_p(0)$ are determined by minimizing $\chi^2$ of Rydberg–perturber bound state energies for $n = 30$, $n = 33$, and $n = 36$ that are experimentally available [15].

Experimental bound state energies are taken from Figure 1. of DeSalvo et al. [15] as the locations of relative atom number minima on the best fit curves. Approximate standard deviations are determined from the full width at half maximum of the best fit curve peaks via the relation $\sigma = \frac{\text{FWHM}}{2\sqrt{2\ln(2)}}$, which assumes a Gaussian shape. Meanwhile, to calculate theoretical bound state energies, we use a discrete variable representation (DVR) using a fine radial grid, ensuring a convergence of the bound states better than 0.1%. This is repeated over a range of values of $a_s(0)$ and $a_p(0)$.

A contour map of $\chi^2$ as a function of $a_s(0)$ and $a_p(0)$ with the $1\sigma$ region (a 68% confidence region) labeled is given in Figure 2, following the statistical methods described in Ref. [42]. The minimum and maximum values of $a_s(0)$ and $a_p(0)$ within this region are used for determining their uncertainties. Thus, we obtain $a_s(0) = -13.135 \pm 0.035$ a$_0$ and $a_p(0) = 9.11 \pm 0.12$ a$_0$. The value for the *s*-wave is close to the previously determined value $a_s(0) = -13.2$ a$_0$. On the contrary, for the *p*-wave, we observe a larger discrepancy of $a_p(0) = 8.4$ a$_0$. It is worth emphasizing that our results are obtained, including the core–neutral interaction. In contrast, these were not included in the work of DeSalvo et al. [15]. Very similar results to the ones reported here have been obtained in Ref. [43], confirming the role of $V_{c-p}$ on the value of the effective scattering lengths.

With these scattering lengths, we plot the potential due to a single perturber $V(r) = V_{ea}(r) + V_{ca}(r)$ as a function of interatomic distance in Figure 3. Due to the dominant *s*-wave scattering term, the shape of the potential closely resembles that of $|\psi(r)|^2$ for the Rydberg electron. It is worth noticing that our potentials are very similar to the previously reported ones in Ref. [15]. However, in our case, we include the effect of the ionic core on the neutral atom.

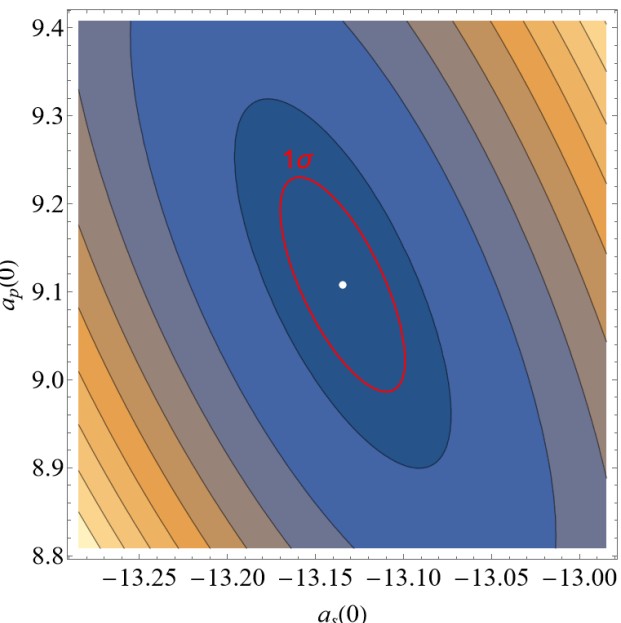

**Figure 2.** A contour map is plotted of $\chi^2$ for Rydberg–perturber bound state energies as a function of the scattering lengths $a_s(0)$ and $a_p(0)$. The experimental data used are taken from DeSalvo et al. [15]. The $1\sigma$ contour line represents a 68% confidence region, and the white dot at the center minimizes $\chi^2$.

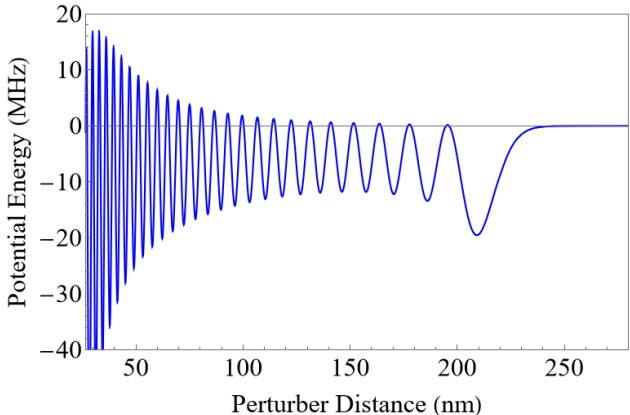

**Figure 3.** Potential energy curve for the $49\,^3S_1 + 5\,^1S$ state of $^{84}$Sr.

### 3.2. Determining BEC Density Parameters

The density distribution $\mathcal{N}(\rho)$ of the BEC is described within the Thomas–Fermi approximation. $\mathcal{N}(\rho)$ depends on the Thomas–Fermi radius of the trap $R_{TF}$, the peak BEC density $\mathcal{N}_{max}$, and the temperature $T$. However, changing $R_{TF}$ only affects $\mathcal{N}(\rho)$ by rescaling $\rho$, which has no effect on the normalized lineshape. Meanwhile, the values of $\mathcal{N}_{max}$ and $T$ are set, with the goal of most closely matching the experimental conditions of Ref. [18]. To this end, $T$ is always adjusted so that the condensate fraction matches the experimental value given in Table 1 of Ref. [19] for the corresponding measured lineshape ($n = 49$, $n = 60$, or $n = 72$). Due to the absence of a second experimentally measured parameter, we can only determine $\mathcal{N}_{max}$ by fitting it to the experimental lineshape. The density distributions $\mathcal{N}_{BEC}(\rho)$ and $\mathcal{N}_{th}(\rho)$ for $n = 49$ are plotted in Figure 4. This figure shows the expected density profile for a BEC in a harmonic trapping potential. We also notice an enhancement of the atomic thermal density around $R_{TF}$. A summary of the values of all relevant parameters is given in Table 2.

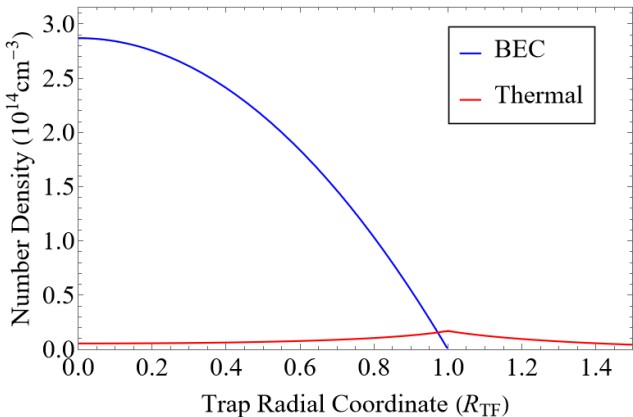

**Figure 4.** Density of the BEC and thermal atoms is plotted as a function of distance from the center of the trap. At the Thomas–Fermi radius of the trap, $R_{TF}$, the BEC density vanishes and the thermal density peaks. The temperature and peak BEC density used to obtain this plot match those used for the $n = 49$ simulations.

**Table 2.** BEC simulation parameters. $T$ is chosen such that each condensate fraction matches the corresponding experimental value from Table 1 of Ref. [19]. $\mathcal{N}_{max}$ is determined by fitting to the experimental lineshape of Ref. [18].

| $n$ | $n^*$ | $T$ (nK) | $\mathcal{N}_{max}$ ($10^{14}$cm$^{-3}$) | $|\Delta\omega_{min}|$ (MHz) |
|---|---|---|---|---|
| 49 | 45.629 | 171.2 | 2.87 | 0.005 |
| 60 | 56.629 | 187.0 | 3.65 | 0.003 |
| 72 | 68.629 | 170.2 | 3.56 | 0.002 |

*3.3. Accuracy of the Quasi-Static Simulations*

The resultant simulated lineshapes are plotted in Figure 5 alongside experimental data [18] for $n = 49$, $n = 60$, and $n = 72$, showing good agreement. There are two significant sources of uncertainty: random error and effective scattering length uncertainty.

1. To determine the random error, 10 lineshapes $A1$–10 of $10^5$ excitations each are generated for each $n$. The average of lineshapes $A1$–10 gives lineshape $A$, and their standard deviation gives the random error.

2. To infer uncertainty due to the intrinsic errors attached to the effective scattering lengths, two lineshapes $B1$ and $B2$ of $10^6$ excitations each were generated for each $n$, calculating detunings using the scattering length lower and upper bounds, respectively. Both $B1$ and $B2$ used the same perturber arrangements as the $10(10^5) = 10^6$ excitations of lineshape $A$. Two new lineshapes, $B_{min}$ and $B_{max}$, are defined as $B_{min}(\nu) = \min\{B1(\nu), B2(\nu)\}$ and $B_{max}(\nu) = \max\{B1(\nu), B2(\nu)\}$ without being normalized. Then $A - B_{min}$, wherever positive, is an additional source of lower uncertainty, and $B_{max} - A$, wherever positive, is an additional source of upper uncertainty.

The final lineshape denoted as $A$ includes both sources of uncertainty added in quadrature.

The most noticeable discrepancy between simulation and experiment is the height of the absorption peak at zero detuning. Based on the $|\Delta\omega_{min}|$ values from Table 2, the quasi-static approximation fails for very small detunings, as Equation (2) dictates, and it is depicted in Figure 6. In particular, for the largest principal quantum number considered here, our lineshape simulations based on the quasi-static approximation should be valid up to detunings ~2 kHz. However, this could only affect the lineshape's details very close to zero detuning, not the height of the entire peak. It is also possible that we are underestimating the number of thermal atoms, which, as we will see, are responsible for the peak at zero detuning. However, the most straightforward explanation in our view is that the peak height is very sensitive to the bandwidth of the light, and the experimental bandwidth may have been smaller than 1 MHz. A smaller bandwidth would make the

absorption peak narrower and taller, better fitting the experimental data. On the positive side, our simulations capture well the blue detuned region of the lineshape in contrast to previous simulations [19].

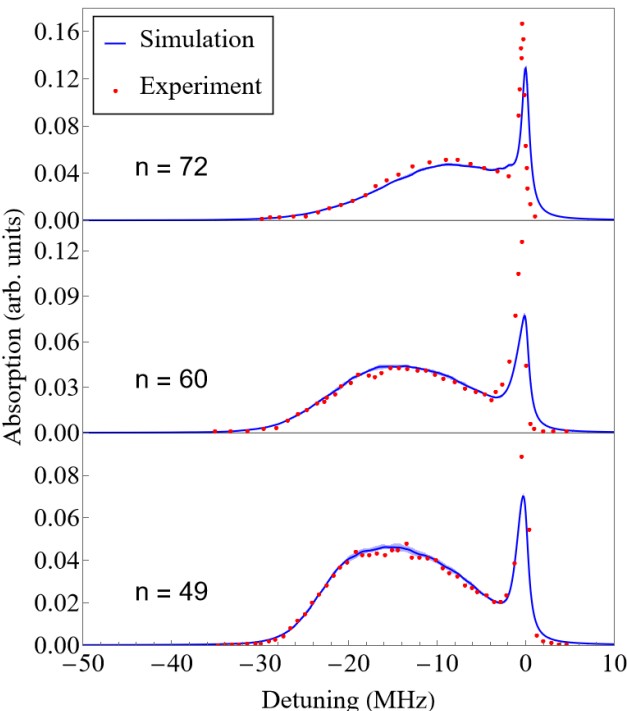

**Figure 5.** Simulated lineshapes for Rydberg excitations to the $49^3S$, $60^3S$, and $72^3S$ states are plotted together with a representative subset of experimental values from Camargo et al. [18]. Both simulated and experimental lineshapes are normalized to 1. Simulation uncertainty due to the MC sampling is shown as the shaded blue area around the blue solid line. The uncertainty is very small and may not be visible everywhere.

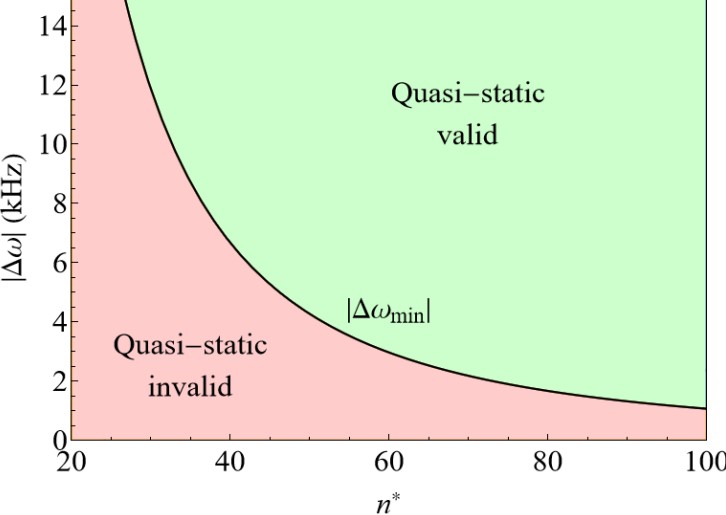

**Figure 6.** Validity of the quasi-static approximation for Rydberg excitations in an $^{84}$Sr BEC based on Equation (2). The green region denotes the region of the parameter space (detuning and effective principal quantum number) in which the quasi-static approximation is accurate. In contrast, the red region denotes where quasi-static approximation cannot be applied.

A more significant discrepancy is the difference in shape between simulation and experiment for $n = 72$. The clearest reason for discrepancy is the use of approximate

effective scattering lengths with first order perturbation theory in our simulations. Effective scattering lengths were obtained via fitting of bound states with $n$ in the range 30–36, but as $n$ increases beyond 36, higher semi-classical momenta $k$ become more prevalent, so our momentum-dependent effective scattering lengths may become inaccurate. If this is indeed the reason for the discrepancy, then the lineshapes suggest that the inaccuracy starts to become prominent between $n = 60$ and $n = 72$.

### 3.4. Roles of the Rydberg Core–Perturber Interaction and the Thermal Fraction

Several variations of simulated lineshapes are plotted in Figure 7 alongside experimental data [18] for $n = 49$, $n = 60$, and $n = 72$. Note the logarithmic scaling. Uncertainty was calculated following the same procedure as before. These plots allow us to understand the effects of the core–perturber interaction $V_{c-p}$ and the thermal atoms on the lineshape better.

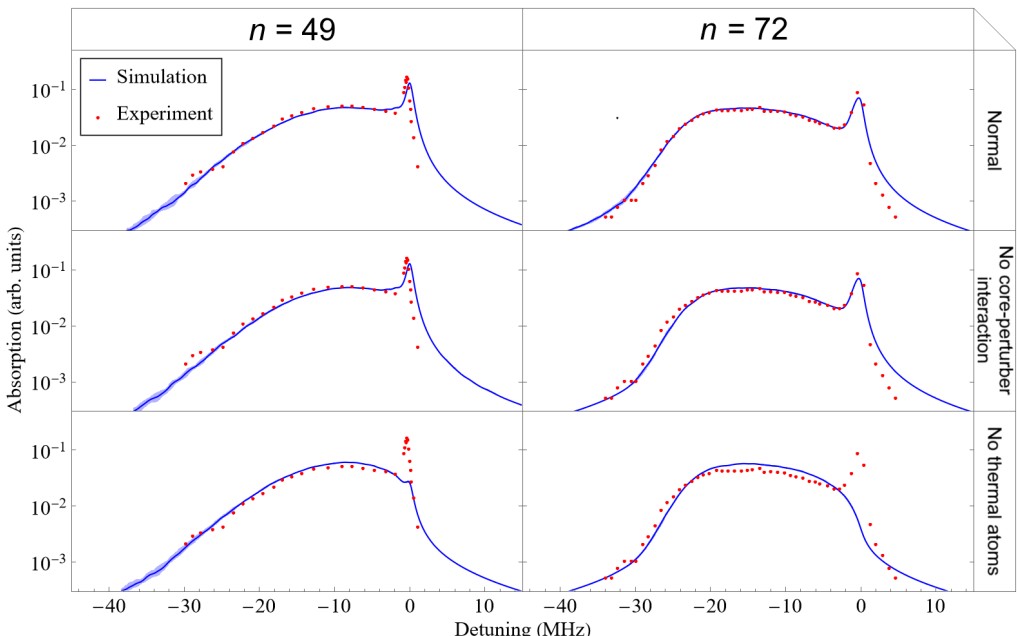

**Figure 7.** Several variations of simulated lineshapes for Rydberg excitations to the $49s$ and $72s$ states are plotted together with a representative subset of experimental values from Camargo et al. [18]. Both simulated and experimental lineshapes are normalized to 1. Simulation uncertainty is small but illustrated. The second row excludes the effect of the core–perturber interaction $V_{c-p}$. The third row uses a condensate fraction of 1 so that there are no thermal atoms.

Clearly the thermal atoms are responsible for the absorption peak at zero detuning. Most thermal atoms reside in low-density regions outside $R_{TF}$, where it is very likely that there are no nearby perturbers. Meanwhile, the core–perturber interaction slightly lengthens the tail of the lineshape. The effect is small because $V_{c-p}$ is negligible except at very small interatomic distances. However, when perturbers are sufficiently close to the Rydberg core, $V_{c-p}$ causes the detuning to be more negative. Therefore, the core–perturber interaction is responsible for the wonderful agreement between our simulations and the experimental lineshape at medium-large detunings.

For Rydberg excitations at realistic densities, the electron–perturber interaction always controls the overall shape of the absorption spectrum, whereas the core–perturber interaction is a relatively minor effect. On the other hand, the core–perturber interaction controls the lifetime of a Rydberg atom due to chemi-ionization processes [20,44–46]. However, at sufficiently high densities for a given principal quantum number, the core–perturber interaction may begin to significantly affect the Rydberg excitation lineshape, as is shown in Figure 8.

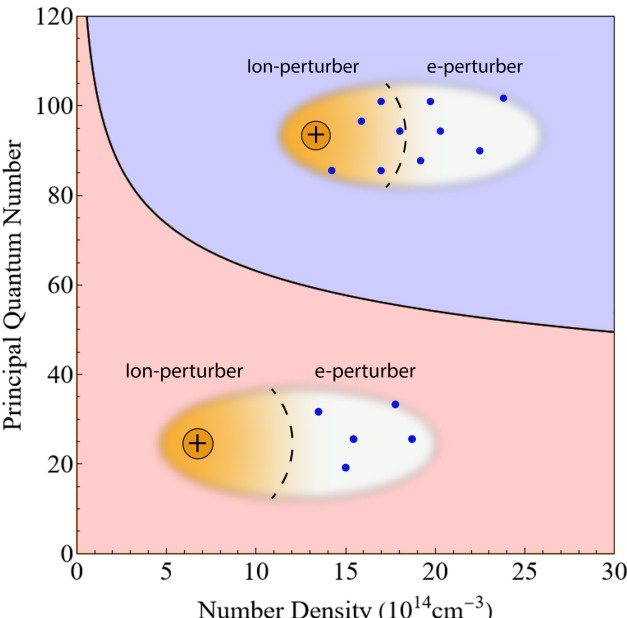

**Figure 8.** Phase space for Rydberg excitations. In the red region, the electron–perturber interaction most likely dominates the core–perturber interaction for all perturbers. On the other hand, in the blue region, the core–perturber interaction most likely dominates the electron–perturber interaction for at least one perturber (the nearest neighbor).

The core–perturber interaction strengthens as interatomic distance decreases. Hence, the red region represents the dominance of the electron–perturber interaction not just for the nearest neighbor but for all perturbers, whereas the blue region denotes the range of density and principal quantum numbers such that the core–perturber interaction dominates over the electron–perturber interaction for the nearest neighbor. These regimes would explain why the core–perturber interaction does not play an essential role in the lineshapes studied. However, it is worth noticing that these regimes are biased in favor of the core–perturber interaction, which is much stronger at shorter distances. As a result, the separatrix between those regimes should be considered as a guide more than an actual separatrix. Indeed, in real systems, that line will describe a transition area rather than a separatrix.

On the other hand, these regimes could be helpful to understand the role of chemi-ionization reactions, since the larger the probability of finding a perturber close to the core, the higher the reaction probability is. For instance, and surprisingly enough, despite the absence of a *p*-wave shape resonance between the electron and perturber, the $^{84}$Sr decay time shows a threshold behavior around $n \sim 80$ [21], very similar to the observations in Rb [20]. Moreover, this behavior aligns with our simulations, showing a transition from an electron–perturber- to a core–perturber-dominated interaction at a principal quantum number similar to 80. Indeed, a more detailed study of this phenomenon will be published elsewhere.

## 4. Summary and Conclusions

This work comprehensively describes the quasi-static lineshape theory applied to Rydberg excitations in high-density media. In particular, we provide a systematic approach to treat lineshapes of Rydberg excitations using effective *s*-wave and *p*-wave scattering lengths and a complete Rydberg–perturber interaction. The method has been tested against $^{84}$Sr Rydberg excitation due to the extensive experimental data and its relevance in Rydberg polaronic physics. Our results show a remarkable agreement for the lineshape, particularly in the mid-to-large detuning range and the blue-detuned side of the lineshape, which has not been achieved before for the system under consideration.

Our method has proven to be a valuable tool for assessing the role of the thermal atoms on the lineshape, which are dominant at low detuning. Similarly, we have explored

the limitations of our approach, finding that it is valid in all detuning ranges except for smaller ones. Finally, we have investigated the role of core–perturber interactions, finding two regimes: one dominated by electron–perturber interactions characteristic of moderate-density media and the other dominated by core–perturber interactions. The last regime could affect the lineshapes but, more importantly, the Rydberg atom's decay via chemi-ionization processes, which could explain the threshold behavior on the decay lifetime of Rydberg's excitation in high-density media. Therefore, the transition between these two regimes deserves further investigation. Finally, it is worth emphasizing that the present method could be extended toward the treatment of ion–Rydberg systems [47–50].

**Author Contributions:** J.P.-R. conceptualized and supervised the project; T.S. programmed the simulations and created the figures. Both authors devised the methodology, analyzed the data, and wrote and edited the manuscript. All authors have read and agreed to the published version of the manuscript.

**Funding:** This material is based upon work supported by the National Science Foundation under Grant No. NSF PHY-1852143. J.P.-R. acknowledges the support of the Simons Foundation.

**Data Availability Statement:** The data presented in this study are available on request to the authors.

**Acknowledgments:** We acknowledge T. Killian and B. Dunning for useful discussions.

**Conflicts of Interest:** The authors declare no conflict of interest.

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
