# Peer review of "Quasi-Static Lineshape Theory for Rydberg Excitations in High-Density Media"

_atoms, doi:10.3390/atoms11060095_

Round 1

Reviewer 1 Report

Referee report

On manuscript “Quasi-static Lineshape Theory for Rydberg Excitations” by T. Scheuing and J. Pérez-Ríos

 The paper is a theoretical study of the lineshape of the Rydberg excitations in cold matter. The authors develop a statistical approach to the description of the influence of the nearby perturbers on the shape of the line corresponding to Rydberg states of strontium. Two perturbation mechanisms are considered: the weakly-bound electron – perturber interaction, and atomic core – perturber interaction. The former is described using effective scattering length approach, while the latter is assumed to be dominated by polarization interaction. The method developed is quite intuitive and scientifically sound. The results obtained for excitation to 493S, 603S, and 723S states are compared with recent experimental data, and remarkable agreement is demonstrated. The field of the study is currently a hot topic so I expect the paper will be of great interest to the readers of Atoms. I believe the paper is, in principle, suitable for publication in Atoms. However, I have small comments that should be addressed before the paper can be recommended for publication:

i)             In the description of the approach realization the authors say (lines 130-132): “We choose the distance r of each perturber from the Rydberg core. Since the nearby density is constant, this is a uniform distribution in space, or one proportional to r2….”

 This is correct for excitations to Rydberg nS states, which the authors illustrate their method with. However, for states with higher angular momenta one needs to consider the angular features of the Rydberg electron density. Nevertheless, on lines 143-144 the authors say that “This computational approach is applicable to BEC’s in different trap geometries and properties, as well as to different Rydberg states. Thus, it is fully general.” This requires a clarification. It is obvious that the approach can be easily generalized to Rydberg states with different angular momenta: the distribution can be altered to account for the shape of the specific Rydberg orbital, and then an averaging will need to be done over the orientation of the orbital.

 ii)           The authors assume that the density of the media is constant near the Rydberg excitation. This is correct for Rydberg states with principal quantum numbers which are not very high. Otherwise, as the authors note on lines 123-125, the radius of the Rydberg orbital increases as a square of the effective principal quantum number and will eventually become large enough for the variation of the media density to be significant. The authors should provide some estimates of the upper limit of the effective principal quantum number their theory is still valid for. This also concerns the validity of the neglecting the trap geometry.

iii)         When discussing effective scattering length on lines 93-95 and 154-157, the authors only cite Ref. [15]. While indeed the effective scattering length was used in Ref. [15], the concept was introduced before (see, for example, [https://journals.aps.org/pra/abstract/10.1103/PhysRevA.66.013403]).

iv)          In the context of the discussion of the roles of core-perturber and electron-perturber interaction it seems reasonable to cite works [https://iopscience.iop.org/article/10.1088/0953-4075/24/8/015 , https://iopscience.iop.org/article/10.1088/0953-4075/24/8/016].

v)           Shaded blue areas in Fig. 5 are barely visible. Perhaps they should be plotted in different color, or magnified in an inset.

 Recommendation: the paper can be accepted for publication after the authors amended the manuscript in accordance with the comments above.

Reviewer 2 Report

The manuscript "Quasi-static Lineshape Theory for Rydberg Excitations" have developed a general quasi-static lineshape theory for Rydberg excitations in high density media. Rydberg-perturber bound states gives rise to interesting many-body effects. Rydberg’s properties in high density media characterize electron-neutral interactions, and estimate hard-to-measure physical properties of the background gas. However, there is no general theoretical approach to explain the Rydberg excitation lineshape. Therefore, the work of this manuscript is very meaningful. They simulated lineshape of Rydberg excitation and the good agreement with the Sr Bose-Einstein condensate (BEC) experiment proves the reliability of their established method. I think the manuscript suitable for publication in Atoms.
However, I have a comment that should be addressed before the manuscript can be recommended for publication: Noted that the high-density media is occurred in an atom trap containing a Bose-Einstein condensate (BEC). The title and abstract of the manuscript do not highlight the characteristics of the work is Rydberg’s properties in high density media. This is very crucial. Otherwise, the paragraph "...we discuss the role of the thermal atoms and core-perturber interactions, generally disregarded in Rydberg physics..." in the abstract would be considered a mistake in the field of calculating the Rydberg energy levels of free atoms, as the current calculations for Rydberg spectroscopy of free atoms consider the influence of Rydberg electrons on the core [can be seen in the references: M. Aymar, C. H. Greene, and E. Luc-Koenig, Multichannel Rydberg spectroscopy of complex atoms, Rev. Mord. Phys.68,1015(1996)].

Reviewer 3 Report

The paper calculates lineshapes of Rydberg states in BEC gas of Sr atoms in the quasistatic  approximation. There were previous calculations for this specific system, Ref. [15], and the present calculations don't produce significantly new results, except perhaps an investigation of the perturber-core interaction. However, this interaction affects very little the lineshapes.

The authors calculate the broadening in the spirit of the Fermi approach, developed later by Omont, whereby the width is expressed in terms of the scattering length or an effective scattering length. However, more sophisticated theories developed later express the broadening in terms of scattering phase shifts and collision rates including an arbitrary number of partial waves, see, for example Alekseev V A and Sobel’man I I 1966 Sov. Phys.-JETP 22 882-8. Why do the authors ignore this, more accurate, approach?

p. 1, line 29:

"On the contrary, when such a resonance exists, a quasi-static approach for the lineshape leads to a proper description of the Rydberg excitation lineshape [25]. Therefore developing a proper framework for the theory of Rydberg excitation spectra is necessary." The motivation is unclear: if the theory does exist, why a new development is necessary?

p. 2: why is excitation time defined as 1/|Delta\omega|? It leads to the infinite excitation time in the absence of perturbation which is very strange, I would define the excitation time as 1/(transition rate).

p. 5 why is so-called "p-wave scattering length" estimated to be zero? The authors refer to an experimental paper, but for a theoretical paper a theoretical justification is necessary. Later on the authors give a_p=9.11 obtained from their statistical analysis, but this is also from experimental results. Aren't ab initio e-Sr scattering calculations possible? From all discussions on p. 5 it seems that the "p-wave scattering length" is an ambiguous concept, and the theory should be reformulated in terms of the scattering phase shifts as functions of energy.

Fig. 4: what are the units on horizontal axis? It is doubtful these are a.u.

Sec. 3.3: the results in Fig. 5 were obtained by fitting the theoretical parameters (effective scattering lengths) to experimental values. So "the good agreement" with experiment should not be surprising. This conclusion is confirmed by the fact that "agreement" is much better for n=33-36 (values for which the fitting was done) and worse for n=60 and 72 (p. 9). It is not clear to the reader what is the purpose of the whole exercise. To fit "effective scattering lengths" to the experimentally observed line shapes in the region n=33-36?

Fig. 5: "Simulation uncertainty due to the MC sampling is shown as the shaded blue are around the blue solid line." I don't see the shaded blue in this figure. Perhaps the MC uncertainty is so small that it should be depicted  by some other means.

In conclusion it is hard to see if the paper contains new physics. The present version of the lineshape theory for Rydberg states in BEC gases was developed earlier in  [18,19,25]. Perhaps the authors can rewrite the manuscript emphasizing on the new physics results. Then the paper can be reconsidered for publication.
